

# MTSv: rapid alignment-based taxonomic classification and high-confidence metagenomic analysis

Tara N. Furstenau[1], Tsosie Schneider[1], Isaac Shaffer[1],
Adam J. Vazquez[2], Jason Sahl[2] and Viacheslav Fofanov[1,2]

[1] School of Informatics, Computing, and Cyber Systems, Northern Arizona University, Flagstaff, Arizona, United States
[2] Pathogen and Microbiome Institute, Northern Arizona University, Flagstaff, Arizona, United States

## ABSTRACT

As the size of reference sequence databases and high-throughput sequencing datasets continue to grow, it is becoming computationally infeasible to use traditional alignment to large genome databases for taxonomic classification of metagenomic reads. Exact matching approaches can rapidly assign taxonomy and summarize the composition of microbial communities, but they sacrifice accuracy and can lead to false positives. Full alignment tools provide higher confidence assignments and can assign sequences from genomes that diverge from reference sequences; however, full alignment tools are computationally intensive. To address this, we designed MTSv specifically for alignment-based taxonomic assignment in metagenomic analysis. This tool implements an FM-index assisted q-gram filter and SIMD accelerated Smith-Waterman algorithm to find alignments. However, unlike traditional aligners, MTSv will not attempt to make additional alignments to a TaxID once an alignment of sufficient quality has been found. This improves efficiency when many reference sequences are available per taxon. MTSv was designed to be flexible and can be modified to run on either memory or processor constrained systems. Although MTSv cannot compete with the speeds of exact k-mer matching approaches, it is reasonably fast and has higher precision than popular exact matching approaches. Because MTSv performs a full alignment it can classify reads even when the genomes share low similarity with reference sequences and provides a tool for high confidence pathogen detection with low off-target assignments to near neighbor species.

# INTRODUCTION

Advancements in sequencing technologies over the last few decades have transformed the way that we study complex microbial communities, which in turn has had a large impact on human health, ecology, and industry. In the past, characterization of microbial communities was restricted to sequencing the few individual members that could be successfully isolated and cultured. Fortunately, due to dramatic improvements in the efficiency and cost-effectiveness of sequencing technologies, we can now use shotgun

Corresponding author
Tara N. Furstenau,
tara.furstenau@nau.edu

metagenomic sequencing to study the composition of entire microbial communities in a culture-independent manner.

Unlike traditional genomic sequencing, which uses DNA isolated from a single organism, metagenomic sequencing is performed on the full complement of DNA captured from a mixture of organisms. Many different methods can be used to characterize metagenomic samples including reference-based and reference-free approaches. Here we focus on reference-based taxonomic classifiers which assign sequencing reads to reference taxa in large sequence databases. Initially, this was accomplished by repurposing existing traditional rapid alignment tools like BLAST (*Altschul et al., 1990*; *Huson et al., 2007*; *Camacho et al., 2009*; *Bazinet et al., 2018*), and Bowtie2 (*Langmead & Salzberg, 2012*; *Segata et al., 2012*). Traditional read aligners (*e.g.*, Bowtie2) use various heuristics to rapidly search for good (although not necessarily optimal) alignment locations within a single reference genome. Although there may be many potential alignment locations within the genome, aligners will typically perform a prescribed level of search effort and then report only the best alignment found (or in cases of a tie, randomly pick one) (*Langmead & Salzberg, 2012*). This is appropriate for the application that they were designed for, but it presents a problem when attempting to find alignments against large multi-species reference databases. The problem is that the level of search effort may be expended before all references are checked thereby missing equally good or better alignments to certain taxa. Reference databases already vary widely in the number of sequences per taxa, and this would further bias assignments to taxa that are overrepresented in the database.

BLAST and its many variations (*Altschul et al., 1990*; *Morgulis et al., 2008*; *Camacho et al., 2009*) were designed to rapidly search large sequence databases and can report all alignments over a certain quality. This reduces bias by performing a more exhaustive search but requires many more computationally expensive alignments and is too slow to feasibly process large datasets (*Breitwieser, Lu & Salzberg, 2019*). Similarly, Bowtie2 provides a mode that searches for and reports all valid alignments, but this increases the run time from minutes to days when there are many similar sequences in the database. As databases and sequencing throughput continue to grow, such approaches will become increasingly impractical.

Several approaches have been introduced to handle the growing size of reference databases used for metagenomic analysis. Due to the higher levels of conservation of protein compared to DNA, protein-based aligners like DIAMOND (*Buchfink, Xie & Huson, 2015*; *Buchfink, Reuter & Drost, 2021*), MMSeqs2 (*Steinegger & Söding, 2017*; *Mirdita et al., 2021*), and Kaiju (*Menzel, Ng & Krogh, 2016*) take advantage of non-redundant protein databases and are orders of magnitude faster than BLAST (*Ye et al., 2019*). However, protein-based tools tend to classify fewer reads (because the databases contain only coding regions) and they have higher rates of misclassification (*Ye et al., 2019*). Centrifuge introduced a compression method which reduces the number of redundant reference sequences that must be searched (*Kim et al., 2016*). However, creating the compressed dataset is computationally intensive and thus difficult to update and customize and users typically rely on versions maintained by the developers.

The lossy compression of the database also results in lower sensitivity (*Ye et al., 2019*). MetaPhlan (*Segata et al., 2012*) uses Bowtie2 for alignments but restricts alignments to a set of marker genes that are specific to certain taxa. This approach is very quick, but resolution is low and unevenly distributed across taxa (*Ye et al., 2019*). Metalign (*LaPierre et al., 2020*) reduces the number of reference genomes by filtering out sequences that are not likely to be in the dataset. It uses a MinHash-based approach (*Koslicki & Zabeti, 2019*) (like Mash (*Ondov et al., 2016*) and Sourmash (*Brown & Irber, 2016*; *Pierce et al., 2019*)) to identify reference genomes that share a high percent of k-mers with sample sequences and only includes these genomes in the reference set. With this approach, some species are likely to be left out of the filtered database by mistake (*e.g.*, species that diverge from the reference sequences and low abundance organisms) and Metalign only includes a single representative genome per taxon which may reduce assignments. Metalign uses Minimap2 (*Li, 2018*) to align reads to the filtered database. Minimap2, like other traditional read aligners, was not designed to work with multi-species reference databases and as we show here, it can introduce biases in assignments.

Newer methods, based on exact k-mer matching rather than full alignment, have been introduced (*e.g.*, Kraken2; *Wood & Salzberg, 2014*; *Wood, Lu & Langmead, 2019*) and provide significant speed-ups. However, the heuristics can have large impacts on the accuracy of the results. For example, the selection of a fixed k-mer size requires a fine balance between sensitivity and specificity; longer k-mers will be more specific to a particular taxon but they are less likely to have exact matches due to errors or genetic variation (*Lu et al., 2017*). These tradeoffs work reasonably well in many but not all cases and can lead to low resolution of certain taxa and/or missed assignments when sequences diverge from the references.

Many situations call for higher accuracy and higher confidence in taxonomic assignments and there is a need for a tool that can perform full alignments of metagenomic reads at a reasonable speed. These situations are generally those in which the presence or absence of specific species needs to be established with high confidence. These cases include the detection of organisms known to be associated with clinical syndromes (*Chiu & Miller, 2019*; *Gu, Miller & Chiu, 2019*), public health surveillance efforts (*Miller et al., 2013*), forensic evidence (*Kumari et al., 2022*; *Clarke et al., 2017*; *Robinson et al., 2021*), or in bio threat (*Bazinet et al., 2018*; *Karlsson et al., 2013*; *Minogue et al., 2019*) or biocontamination (*De Filippis et al., 2021*; *Wood et al., 2021*) situations. In such cases, a tendency toward false positive assignments can be highly detrimental as independent validation may be needed either through experimentation or often by manually aligning reads to the reference genome of the suspected organism. In such cases, it may be less time consuming to begin with alignment-based taxonomic assignments rather than attempt to validate the results from faster, yet less accurate tools.

Here we introduce a new alignment-based taxonomic assignment algorithm called MTSv (https://github.com/FofanovLab/mtsv_tools). Our goal in developing MTSv was to use efficient data structures and algorithms, like those used by traditional read alignment tools (*e.g.*, Bowtie2), in a way that was specifically designed to handle metagenomic data. As described above, traditional read aligners typically report only

the top alignments for each read which causes them to miss assignments to multiple taxa in mixed reference databases. However, when forced to report all valid alignments, they are very slow because once they find a valid alignment for a taxon, they continue the exhaustive search across all reference sequences for the same taxon (this is especially slow when there are many similar reference sequences present in the database which is often the case). For taxonomic assignment however, the location and number of potential alignments is not relevant; it is sufficient to know that there is at least one high-quality alignment to a given taxon. With this in mind, we designed our algorithm to bypass further alignment attempts once an alignment of sufficient quality is found resulting in highly accurate assignments with significant speedups. Using both simulated and real datasets, we demonstrate that our full alignment approach (1) offers unbiased assignments and is faster than traditional read aligners (2) drastically reduces false positive assignments compared to metagenomic tools that do not provide full alignments (3) increases the number of assignments for taxa that share low similarity with reference sequences, and (4) provides high confidence detection of pathogens while avoiding off-target assignments to near-neighbor species.

# IMPLEMENTATION

## Building the metagenomic FM-index

MTSv implements a custom metagenomic index (MG-index) based on the FM-index data structure (*Ferragina & Manzini, 2000*). To build the index, MTSv takes a multi-FASTA file with headers indicating the Taxonomic ID (TaxID) for each reference sequence (Fig. 1). The reference sequences can be sourced from any DNA sequence collection (*i.e.*, GenBank, RefSeq, etc.) and customized to fit the project. Because MTSv was designed to be highly parallelizable, we recommend building multiple indices from smaller chunks of the reference sequences. This helps reduce the memory requirements and allows for faster processing for both index building and assignment. A tool for splitting up the reference files is included with MTSv.

During the index build, MTSv concatenates the sequences in the FASTA file while recording the location of sequence boundaries and the TaxID associated with each sequence. MTSv then builds a suffix array (*Manber & Myers, 1993*), Burrows-Wheeler Transform (*Burrows & Wheeler, 1994*) and FM-index (*Ferragina & Manzini, 2000*) from the concatenated sequences using the Rust-Bio v0.39.1 package (*Köster, 2016*). The FM-index with the associated sequence metadata constitutes the MG-index. One MG-index is created per FASTA file, and new indices can be added as the reference collection grows without needing to rebuild any of the existing indices.

## Taxonomic assignment

When classifying metagenomic sequences, MTSv uses the MG-Index to narrow down reference sequence locations that are most likely to contain a good alignment. The binning step begins by extracting overlapping substrings of the same size (seed-size) using a user-provided offset (seed-interval) from each read and its reverse complement (Fig. 2). It then uses the MG-Index to search for exact, un-gapped matches for each seed. The seed

**Figure 1 Diagram of MTSv index setup and assignment pipeline.** To build the MG-index, reference sequences are combined into a reference FASTA file formatted with the TaxID in the header. MTSv breaks up large reference files into smaller chunks with a user defined size and builds an MG-index from each chunk. For taxonomic assignment, reads are aligned against each of the MG-Indices. The binning outputs from each MG-Index are merged to provide the final taxonomic assignments.

matches are sorted by location and grouped into candidate regions using specified windows. The number of hits per candidate is tallied and any candidate that does not meet the user-specified minimum number of seed hits is filtered out.

The filtering step is based on a q-gram algorithm which defines the minimum number of exact $k$-mer matches (from all $n - k + 1$ overlapping $k$-mers) that can be expected between an $n$-length read and a reference sequence with at most $e$ mismatches. In the worst case, where all mismatches are evenly spaced across the alignment, the minimum number of matching $k$-mers is: $m = (n + 1) - k(e + 1)$ ($m > 0$ when $\left\lfloor \frac{n}{e+1} \right\rfloor > k$) (*Burkhardt et al., 1999*; *Rasmussen, Stoye & Myers, 2006*; *Reinert et al., 2015*; *Ukkonen, 1992*). If only every $l^{th}$ overlapping $k$-mer is used, the minimum number of matching $k$-mers is expected to be $\frac{m}{l}$. Within these parameters, MTSv is highly likely to find all alignments with up to $e$ mismatches but will still find many alignments with more than $e$ mismatches up to the user provided maximum edit distance cutoff.

After the filtering step, the remaining candidate positions are sorted by the number of seed hits in descending order so that the most promising regions are evaluated first. For each candidate region, MTSv extracts the corresponding range from the reference sequence and looks up the TaxID associated with the reference in the MG-index. If the current read has already been successfully aligned to the TaxID associated with the candidate region, no additional alignment is attempted, and the next candidate region is checked. Otherwise, a Single Instruction Multiple Data (SIMD)-accelerated Smith-Waterman alignment (*Zhao et al., 2013*) is performed between the extracted reference sequence and the query sequence (using a scoring of 1 for matches and −1 for mismatches, gap opening, and gap extension). If the alignment score is sufficiently high, there is one final check to determine if the edit distance is less than or equal to the

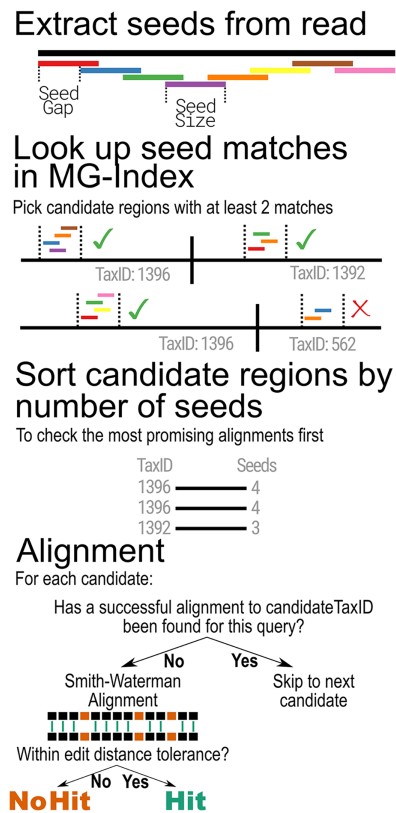

**Figure 2 MTSv alignment algorithm.** Reads are broken into seeds of a user-specified size (seed-size) that are separated by a user-specified offset (seed-interval). Using the MG-index, the positions of exact seed hits are found and recorded. Regions that have at least a user-specified minimum number of seed hits (min-seed) are marked as candidate regions. The candidate regions are first sorted (descending) by the number of seed hits to ensure that the most promising alignments are checked first. The candidate region sequence and the query are aligned using a SIMD-accelerated Smith-Waterman algorithm and the alignment is considered successful if the edit distance of the alignment is less than or equal to the user-specified minimum (edit-rate). Once a successful alignment is found for a TaxID, no further candidates will be aligned for that TaxID.

user-specified edit distance cutoff. If the alignment is considered successful, then no further alignments are attempted for that read against the same TaxID. Skipping all additional alignments to a TaxID avoids many expensive operations and reduces computation time. As the number of reference genomes (and thus the size of a taxonomic bin) continues to rapidly expand, the advantage of this computational speed-up will only become more critical.

Reads are binned against each MG-Index producing separate output files. Each line of the output file contains the read name and a comma separated list of the TaxIDs it aligned to with the edit distance (*e.g.*, "R1:1392=2,1396=3"). The edit distance provided for a TaxID is recorded from the first successful alignment (because no more are attempted afterward) and therefore it is not guaranteed to be the minimum. However, because the alignments start with the most promising candidates first, the edit distance is likely to be near the minimum for each TaxID. Once all output files from the binning step are completed, the files are merged to combine read assignments across multiple MG-Indices.

## MATERIALS AND METHODS

### Comparison to existing software

We compared the performance of MTSv to two traditional read alignment tools (Bowtie2 v2.4.5 (*Langmead & Salzberg, 2012*) and Minimap2 v2.24 (*Li, 2018*)) and two commonly used taxonomic assignment tools designed for metagenomic data (Centrifuge v1.0.4 (*Kim et al., 2016*), and Kraken2 v2.1.2 (*Wood, Lu & Langmead, 2019*)). Bowtie2 is a well-established alignment tool that uses an FM-Index assisted seed-and-extend approach similar to MTSv. Minimap2 was introduced more recently and appears to be a replacement for the BWA suite of tools (*Li, 2007*; *Li & Durbin, 2009*). It uses a hash table index and a seed-chain-align approach that is faster than Bowtie2 and BWA-MEM for Illumina short read alignments (*Li, 2018*). Centrifuge was designed for taxonomic classification of metagenomic sequences and uses an FM-Index data structure. However, unlike MTSv which performs a full alignment, Centrifuge only extends seed hits to find the longest exact matching segments and does not account for gaps or mismatches. Centrifuge assigns scores that favor taxa with longer exact matching segments and then assigns the read to the taxon/taxa with the highest score. Kraken2 is a widely used ultrafast metagenomic classification tool that uses an exact k-mer matching approach. Kraken2 builds a database that maps fixed size k-mers from reference sequences to the lowest common ancestor of all organisms that share the k-mer sequence. The k-mers extracted from sequencing reads can then be looked up in a hash table and assigned to the corresponding TaxID. The final assignment is chosen based on a score from a weighted path classification tree (*Wood & Salzberg, 2014*). Although Centrifuge and Kraken2 (with the help of Bracken (*Lu et al., 2017*)) can use their taxonomic assignments to estimate species abundance profiles, for comparison purposes, we focused only on their performance on taxonomic assignment.

### Building the reference indices

To compare the taxonomic assignment tools, reference indices were built from bacterial/archaeal genome assemblies from RefSeq (*O'Leary et al., 2016*) at the Complete Genome assembly level (121 GB) downloaded on October 28[th], 2019 (Release 96). TaxIDs were assigned to each sequence using the accession to TaxID mapping file provided with the download (all TaxIDs were labeled at the species level or higher) and there was a total of 4,785 unique TaxIDs (4,530 bacteria and 255 archaea). To reduce the memory requirements for building and loading the MG-index, we split the reference file into 13 roughly 10 GB chunks and we built individual MG-indices for each chunk. Centrifuge and Kraken2 do not have an option to split up their data structures, so we built them with a single sequence file using default parameters (Kraken2 k-mer length was 35 bp). For comparison purposes, the Centrifuge index was built without reference sequence compression and to avoid variation due to masking methods, low complexity regions were not masked for any of the tools.

Two additional sets of toy reference indices were built to compare MTSv and the traditional alignment tools. The collection of references was chosen to highlight biases that can be introduced when using traditional alignment tools for taxonomic assignment of

metagenomic reads. The first set contained reference sequences of closely related species in the *Bacillus* genus and was dominated by *Bacillus anthracis* genomes (Table 1). The second was dominated by *Staphylococcus aureus* but also contained a *Staphylococcus epidermidis* reference sequence. The accession numbers for the sequences are available in Supplemental Table S1. MTSv, Bowtie2, and Minimap2 indices were built for both reference sequence collections using default parameters.

## Parameter settings

For Bowtie2, we used parameter settings that were optimized for taxonomic binning of metagenomic reads (*Jaillard et al., 2016*); namely, a seed length of 20 (-L 20), seed error of 0 (-N 0), effort level of 15 consecutive seed extensions (-D 25), and end-to-end mode (—end-to-end–min-score L,-0.6,-0.4). Default values were used for Minimap2. In some cases, the aligners were also run in a way that forced them to find and report all alignments for each read. For Bowtie2 this was achieved using the "all" mode (-a) and for Minimap2 we used the "-N" option to retain up to 5,000 alignments. Our goal was to ensure that the search effort extended enough to cover alignments to all references in the index and the limit of 5,000 was chosen because it was much larger than the number of sequences. Bowtie2 also has a -k method to set an upper bound on the number of alignments but in our example, they were effectively the same. When comparing the performance of these tools they were run on the 2.60 GHz Intel Skylake Xeon nodes on Northern Arizona University's Monsoon high-performance computing cluster with 120 GB of memory using eight threads each.

Default parameters were used for Kraken2 and Centrifuge. Unless otherwise noted, the default parameters were also used for MTSv: the seed size was set to 18 bp, the space between subsequent overlapping seeds was 15 bp, the minimum number of seed matches to perform a full alignment was set to 1, and the maximum edit distance cutoff was 13% or 20 mismatches for 150 bp reads. For MTSv, only TaxIDs with the lowest reported edit distance were considered for the final assignment. For performance comparison these tools were also run using 2.60 GHz Intel Skylake Xeon nodes using 80 GB of memory and eight threads each.

## Performance evaluation using CAMI datasets

To evaluate the performance of MTSv compared to Centrifuge and Kraken2, we used the human microbiome datasets produced for the second Critical Assessment of Metagenome Interpretation (CAMI2) challenge (*Meyer et al., 2022*). The simulated datasets were modelled on Human Microbiome Project Consortium (*Huttenhower et al., 2012*) profiles from five different body sites using NCBI RefSeq (*O'Leary et al., 2016*) complete genome sequences as genome sources (Downloaded on July 8, 2017). We analyzed the short-read data (Illumina HiSeq2000 150 bp paired-end reads) from the gastrointestinal tract, oral cavity, airway, skin, and urogenital tract. The ground truth source of each read was provided, and the performance metrics were calculated using Amber v.2.0.2 (*Meyer et al., 2018*). For both MTSv and Centrifuge, reads with assignments to multiple

taxa were rolled up to the lowest common ancestor using ETE3 v3.1.2 (*Huerta-Cepas, Serra & Bork, 2016*) whereas Kraken2 does this by default.

Amber is a tool that helps standardize comparative assessment of metagenomic binning tools by providing performance metrics evaluated against the ground truth assignments for each read. Amber defines true positives (TP) as sequences that are assigned to the correct taxon at a given taxonomic level and false positives (FP) as those assigned to an incorrect taxon. False negatives (FN) include unassigned sequences and those that are not assigned at the current taxonomic level (*e.g.*, when multiple assignments are classified at a higher common taxonomic level than the one being evaluated). Sample purity (or precision) is the sum of all true positive sequences in the sample divided by the total number of assigned sequences.

$$p = \frac{TP}{TP + FP}$$

Average purity is the precision calculated for each taxonomic bin averaged across all bins. For average purity small bins have the same weight as larger bins whereas larger bins contribute more to the sample purity value. Sample accuracy was calculated as the sum of true positive sequences divided by the total number of sequences (including those that were unassigned).

$$a = \frac{TP}{FN + TP + FP}$$

The average completeness is the sensitivity of each taxonomic bin averaged across all bins. Completeness is calculated as the number of true positive sequences in the bin divided by the true size of the bin. Again, the average completeness weighs all bins equally, regardless of their size.

$$c = \frac{TP}{TP + FN}$$

### Simulating reads

We used Mason v2.0.9 (*Holtgrewe, 2010*) to simulate 150 bp Illumina reads using default parameters. To compare the traditional alignment tools using the toy indices, we simulated and combined 1 million reads from a single reference sequence from each species in the index (Table S1). When analyzing how divergent genomes impacted taxonomic assignment, we used Mutation-Simulator v2.0.3 (*Kühl, 2019*) to simulate SNPs, insertions, and deletions in a reference genome for *Faecalibacterium prausnitzii* (GCA_003312465.1) at rates ranging from 0–15% with indels occurring at 1/10 the rate of the SNPs. From these genome sequences we simulated 1 million paired reads using Mason. These reads were then added to the CAMI2 gastrointestinal tract reads which we confirmed did not contain any sequences from *F. prausnitzii*. For benchmarking the taxonomic assignment tools, 10 million reads were simulated from the reference sequences in the bacterial RefSeq database.

### Pathogen spike-in to soil background

Using real sequencing data, we tested the performance of MTSv, Centrifuge, and Kraken2 at detecting pathogen DNA spiked into a complex soil background. We extracted and quantified, *via* qPCR (*Liu et al., 2012*), DNA from strains of *Bacillus anthracis, Francisella tularensis*, and *Yersinia pestis* from the Pathogen and Microbiome Institute's culture collection. The pathogen DNA was then spiked into DNA extracted from pooled environmental soil samples collected in Puerto Rico in 2016. We used qPCR to verify that none of the pathogens were present in the soil background prior to the spike-in (for *B. anthracis*, we used primers and protocols targeting a SNP in the *plcR* gene (*Easterday et al., 2005*), for *F. tularensis* we used the Ft-sp.FTS_0772 assay primers and protocols (*Öhrman et al., 2021*), and for *Y. pestis*, we targeted the *pla* gene using primers and protocols from *Hinnebusch & Schwan (1993)*, see Supplemental Methods). DNA from each pathogen was spiked into separate aliquots of the soil background at a 2:98 ratio (roughly 2% of the final mixture). The mixtures and the soil background were sequenced on an Illumina MiSeq platform (V3, 2 × 300 kit). The reads were quality-filtered using Fastp v0.20.122 and trimmed to 150 bp. The raw reads are available through the NCBI Sequence Read Archive with the following accessions: SAMN31055686 (*B. anthracis* mixture), SAMN31055687 (*Y. pestis* mixture), SAMN31055688 (*F. tularensis* mixture), and SAMN31055689 (soil background).

## RESULTS

### Traditional read alignment tools were not designed or intended for taxonomic assignment against multi-species reference databases

The standard search effort and reporting modes of traditional read alignment tools result in misclassification when used for taxonomic assignment. Bowtie2 and Minimap2 missed thousands of alignments particularly for species that were underrepresented in the reference index (Table 1) or when genomes from closely related species were included. By default, Bowtie2 reports only the best alignment found after a defined level of search effort (choosing one at random if there is a tie) whereas Minimap2 reports up to five alternative alignments per read. When the standard search effort and reporting modes were used, many reads aligned to the dominant and/or closely related taxa rather than the correct taxon. One index contained closely related species (>99% identity in some cases) from the *Bacillus* genus (*Helgason et al., 2000*; *Rasko et al., 2005*) and was dominated by *Bacillus anthracis* references. Due to the similarity between the genomes, valid alignments for reads of one species were often found within other references. In some cases, the search effort was expended before the alignment to the correct genome was found. The other index included references for *S. aureus* and *S. epidermidis* which share many genes in common but the average nucleotide identity among homologous sequences is low around 75% (*Méric et al., 2015*). Despite this, many alignments to *S. epidermidis* were missed because *S. aureus* sequences were highly overrepresented (961:1) resulting in biased alignments. Minimap2 performed better than Bowtie2 because it reported some alternative alignments by default. But in either case, when using a

**Table 1 Standard read alignment approaches provide incorrect and biased assignments when using multi-species references.** We used two different toy reference indices that were each dominated by a single species and simulated 1 million reads from a single reference for each species. Bowtie2 and Minimap2 were run using the standard mode and "all" mode which forced them to search for and report all alignments. The table reports the number (and percent) of missed alignments (reads simulated from a species that did not align back to the same species).

| Species | Index reference sequences | Number of missed assignments | | | | |
| | | MTSv | Bowtie2 v2.4.5 | | Minimap2 v2.24 | |
| | | | Standard | All | Standard | All |
| --- | --- | --- | --- | --- | --- | --- |
| *Bacillus anthracis* | 116 | 0 (0%) | 254 (0.03%) | 0 (0%) | 10 (0.001%) | 5 (0%) |
| *Bacillus cereus* | 1 | 0 (0%) | **190,759 (19%)** | 0 (0%) | **12,082 (1.2%)** | 105 (0.01%) |
| *Bacillus thuringiensis* | 1 | 0 (0%) | **172,807 (17%)** | 0 (0%) | **9,225 (1%)** | 10 (0.001%) |
| *Bacillus subtilis* | 1 | 0 (0%) | 316 (0.03%) | 0 (0%) | 1 (0%) | 1 (0%) |
| *Staphylococcus aureus* | 961 | 0 (0%) | 28 (0%) | 0 (0%) | 0 (0%) | 0 (0%) |
| *Staphylococcus epidermidis* | 1 | 0 (0%) | **18,532 (2%)** | 0 (0%) | **10,791 (1.2%)** | 47 (0%) |

Note:
The values in bold indicate where ~1% or more of the assignments were missed.

multi-species reference index, the level of search effort did not extend to cover all taxa evenly resulting in biases and missed assignments.

When Bowtie2 and Minimap2 were forced to find and report all alignments, the number of missed assignments was reduced but the runtime and memory requirements increased (Fig. 3). Bowtie2 took over 7 h to align 2 million reads against the *Staphylococcus* index compared to just a couple of minutes in standard mode (~300 times longer). Minimap2 was faster than Bowtie2 although it did not report as many alignments (855 M *vs* 986 M), and it failed to find any alignments for over 5,000 reads. Minimap2 took over an hour compared to ~18 min in standard mode (~4 times longer). Minimap2 also required 111 GB of memory when searching for all alignments compared to 7 GB in standard mode. MTSv took on average 12 min to run (~35 times faster than Bowtie2 and 5 times faster than Minimap2) and required 9.5 GB of memory (only 8% of what Minimap2 required) to achieve the same or better results (in terms of correct taxonomic assignments). MTSv does an exhaustive search of all candidate alignments but it is taxon-aware so it will stop attempting alignments against reference sequences for a TaxID that already has a valid match. Consequently, MTSv was able to skip over 1 billion (93%) redundant and computationally expensive alignments while still correctly assigning the *Staphylococcus* reads.

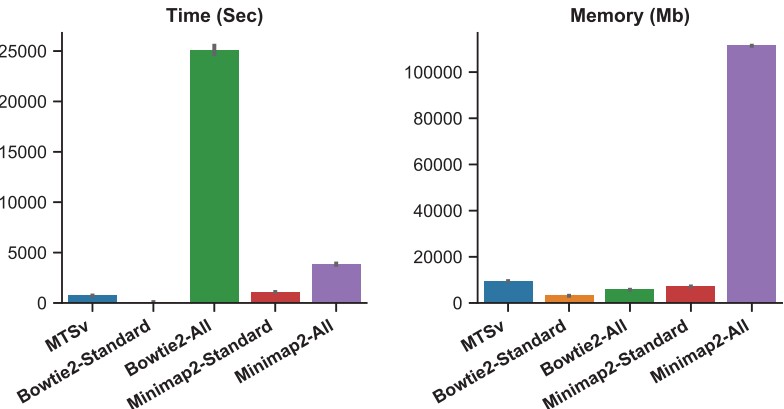

**Figure 3 Forcing traditional alignment tools to report all valid alignments reduced bias when using multi-species references but it increased computation time and memory.** The plots show the run time (left) and memory usage (right) for each alignment tool. Bowtie2 and Minimap2 were run using the standard mode and "all" mode which forced them to search for and report all alignments. These benchmarks were calculated by running two million simulated reads against a toy reference index made up of 961 *Staphylococcus aureus* genomes and one *Staphylococcus epidermidis* genome (Table 1) and are the average of three runs each.

## MTSv has high precision and sensitivity and reduces false positive assignments

MTSv had nearly perfect sample purity and much higher average purity compared to Centrifuge and Kraken2 (Fig. 4, rows 1 and 2, respectively). This indicates that most of the assignments made by MTSv at each taxonomic level were correct whereas, in every case Centrifuge and Kraken2 had incorrect assignments all the way up to the kingdom level (assigning reads to a false archaea bin as indicated by the average purity of 0.5). High precision is the advantage of using a full alignment approach and this is particularly clear when looking at the number of false positive bins (Fig. 5, middle row). At the species level, Centrifuge and Kraken2 assigned reads to thousands of additional taxa that were not in the sample whereas the highest number of false bins for MTSv was 605 for the airways sample. Additionally, the sizes of the false bins were generally smaller for MTSv (Fig. 5, bottom row).

While MTSv has a more conservative approach to making assignments (here the alignment of a 150 bp read could only contain up to 20 mismatches compared to Centrifuge which will report a hit if it finds an exact match >22 bp long) it was still able to assign a large portion of the reads at the species level (Fig. 5, first row). As a result, MTSv maintained high sample accuracy and average completeness (Fig. 4, rows 3 and 4, respectively) by keeping the number of false negatives low. MTSv and Kraken2 had similarly high performance for these metrics while Centrifuge was penalized for having more unassigned reads. The oral sample contained viral sequences that were not included in our bacteria-only reference database, so all the tools had lower average completeness for that sample.

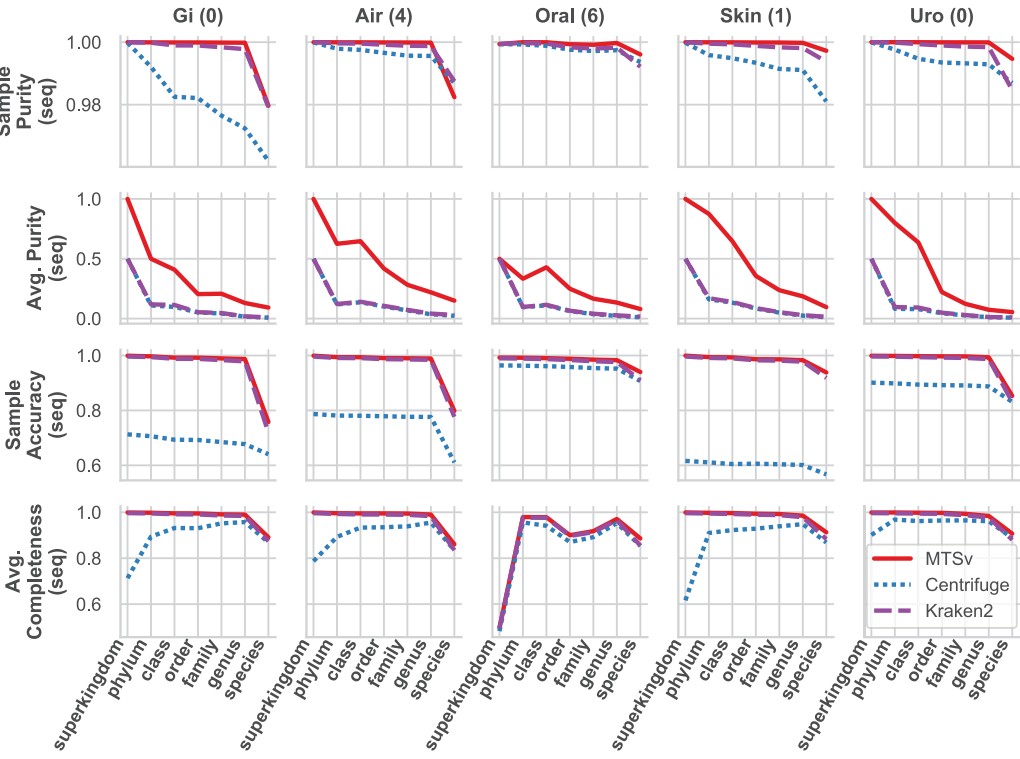

**Figure 4  MTSv generally has high purity, accuracy, and sensitivity when assigning reads with known sources.** The CAMI human microbiome datasets (gastrointestinal, airway, oral cavity, skin, and urogenital) were used to compare the taxonomic binning performance of MTSv, Centrifuge, and Kraken2 at each major taxonomic rank. Sample purity (first row) is the sum of all true positive sequences in the sample divided by the total number of sequences assigned. Average purity (second row) is the precision calculated for each taxonomic bin averaged across all bins. Sample accuracy (third row) is the sum of true positive sequences divided by the total number of sequences (including those that were unassigned). The average completeness (fourth row) is the sensitivity of each taxonomic bin averaged across all bins. For both average purity and average completeness, small bins have the same weight as large bins whereas larger bins contribute greater weight in the sample-based metrics.

## MTSv assigns more reads from genomes that diverge from the reference

The limited number of reference genomes available for taxonomic classification means that sequences in metagenomic samples will often share low similarity with existing database sequences. Divergent sequences contain more gaps and mismatches which will reduce the number of exact k-mer matches. As a result, exact matching approaches tend to miss these assignments completely or have a reduced number of reads assigned to these taxa which can influence downstream abundance estimations.

The number of reads assigned accurately by Centrifuge and Kraken2 to *Faecalibacterium prausnitzii* dropped more rapidly as the genome diverged from the reference sequence (Fig. 6, left). Even at 95% genome similarity, the number of reads assigned dropped to 91% and 94%, respectively, while MTSv was at 98% using default parameters. At 90% similarity, assigned reads dropped to ~50% for Centrifuge and Kraken2 while MTSv still assigned 74%. The reads were spiked into the CAMI2
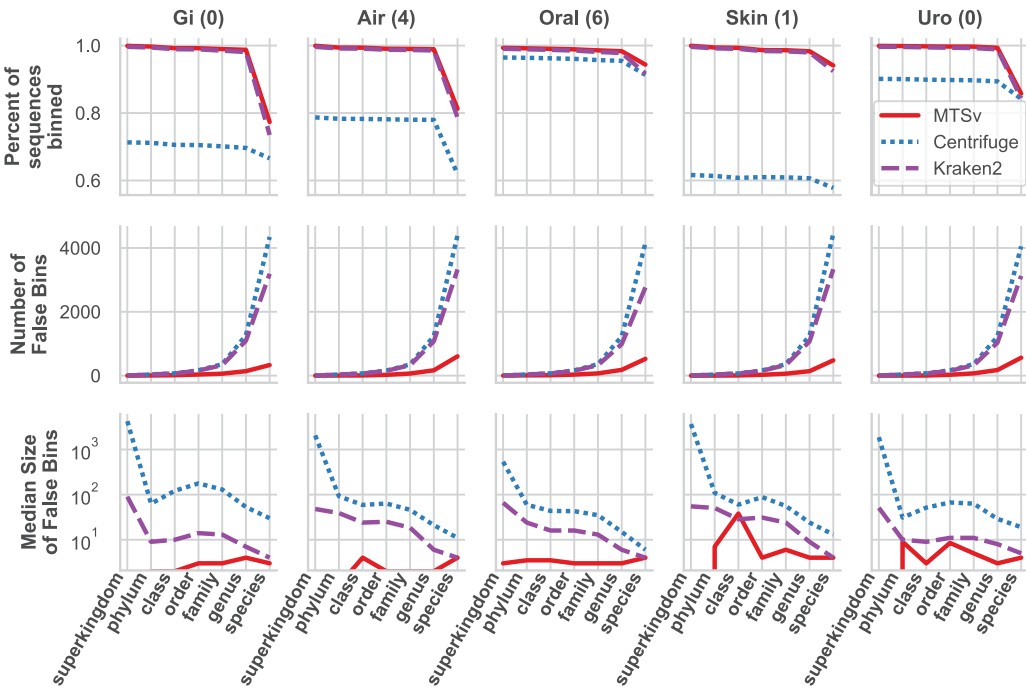

**Figure 5  MTSv classifies a large proportion of reads at the species level and created fewer and smaller false positive bins than both Kraken2 and Centrifuge.** The top row shows the percent of the total reads that were classified at each taxonomic rank for each sample. The second row shows the number of false positive bins assigned by each tool and the bottom row is the median size of the false bins ($\log_{10}$ scale).

gastrointestinal sample to show how the missed assignments affected the proportion of reads assigned to *F. prausnitzii* compared to other species in the sample (Fig. 6, middle). For Centrifuge and Kraken2, the unassigned reads caused *F. prausnitzii* to drop in rank below three other species in relative sequence abundance when genome similarity was 90% and below five species at 85% (Fig. 6, right). Using MTSv default parameters, *F. prausnitzii* dropped below only three species at 85% genome similarity.

With default parameters, MTSv outperformed Centrifuge and Kraken2 in assigning reads from genomes that diverge from reference sequences. By using more sensitive search parameters (seed size = 16, seed interval = 2) and relaxing the edit distance cutoff to 20%, MTSv performed even better. With these settings, MTSv assigned 84% of *F. prausnitzii* reads and read abundance dropped below only one other species when genome similarity was at 85%.

## MTSv provides higher confidence in the detection of pathogens

In microbial forensics and clinical applications, it is important to have a high level of confidence that a detected organism is truly present in the sample. The presence of many false positive assignments can erode this confidence and may require additional time consuming and expensive validation tests. Ideally, adding a single organism to a complex background should not result in many additional false positive assignments. However, although each of the tools was able to detect the correct species in our pathogen spike-in

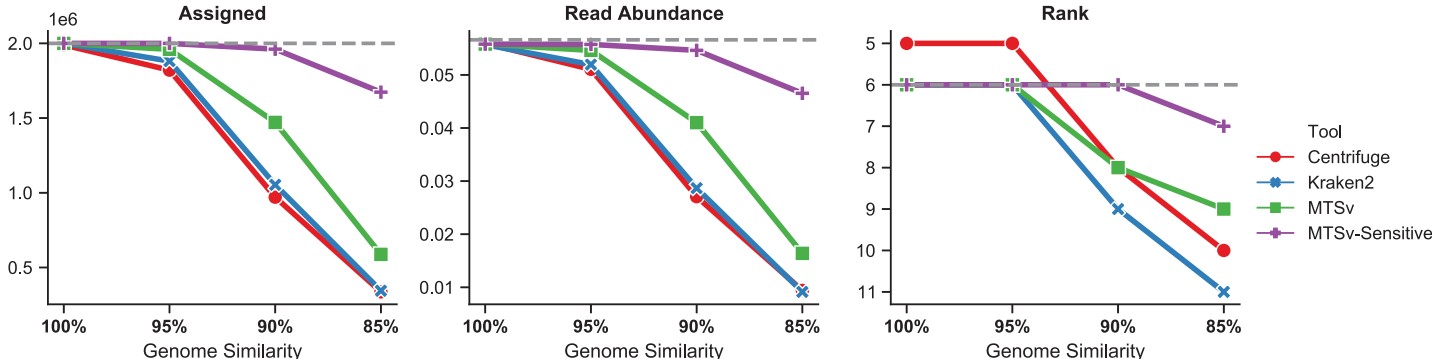

**Figure 6 MTSv assigns more reads from genomes that diverge from reference sequences.** Simulated genomes of *Faecalibacterium prausnitzii* that diverged by 0–15% (100–85% similarity) from the reference sequence were used to simulate two million reads each that were spiked into the CAMI gastrointestinal dataset. The left panel shows the number of reads from *F. prausnitzii* that were assigned correctly (although not necessarily uniquely) by each tool as the genomes diverged. The middle panel is the proportion of reads assigned to *F. prausnitzii* at the species level out of the total number of reads in the CAMI-spikein dataset. The last panel shows how *F. prausnitzii* ranked in read abundance compared to other species in the sample. The grey dashed line in each plot represents the ground truth. MTSv was run using both default and sensitive parameters.

experiments, Centrifuge and Kraken2 introduced many additional false positive assignments to near neighbor species.

The total number of reads assigned by each tool was low which is expected when performing taxonomic assignments of a complex soil mixture against a relatively limited collection of reference sequences (Table 2). Both Centrifuge and Kraken2 assigned more reads than MTSv but many of the assignments had very low support. The median number of k-mers mapped per read for Kraken2 was 19 (16.4%), 14 (12.1%), and 16 (13.8%) for *B. anthracis, F. tularensis,* and *Y. pestis* spike-ins, respectively and the median hit length for Centrifuge was 24 bp for all three samples. In contrast, all the assignments made by MTSv had 20 or fewer mismatches from the reference sequences. Relaxing the mismatch cutoff to 30 increased the number of assigned reads to 706,182, 680,015, and 604,819, respectively, while still requiring an 80% match with the reference sequence.

Despite having fewer total reads assigned, MTSv made the most assignments to the spike-in taxa at the species level. For *B. anthracis* and *Y. pestis,* fewer calls could be made at the species level, likely due to high genetic similarity between these and near neighbor species. For example, *Bacillus anthracis* shares >99% average nucleotide identity with its closest relatives in the *Bacillus cereus* group (*Helgason et al., 2000*; *Rasko et al., 2005*). When reads were assigned to multiple species in a genus, the assignment was correctly made at the genus level. Unlike the other tools, MTSv was less likely to assign reads to near neighbor species within the same genus and introduced fewer new species assignments.

A comparison to the soil background sample (Fig. 7) shows that the additional near neighbor assignments were likely false positives rather than a true signal in the soil background. Because Kraken2 and Centrifuge had many low confidence assignments, we opted to include filtered results for comparison. We used a confidence filter of 0.1 for Kraken2 (*Wood, Lu & Langmead, 2019*; *Ye et al., 2019*) and a score filter of 150 for Centrifuge (*Ammer-Herrmenau et al., 2021*; *Sanderson et al., 2018*); neither filter had a

**Table 2 MTSv assigned the most reads to the spike-in pathogen at the species level and had fewer assignments to near-neighbor species.** The table shows the results from the *B. anthracis*, *F. tularensis*, and *Y. pestis* spike-in samples using MTSv, Centrifuge, and Kraken2. The table shows the number of reads that were assigned uniquely to the spike-in species, the number of reads that aligned to the spike-in genus, the total reads for both the genus and species columns, the number of reads that were uniquely assigned to a non-spike-in species within the genome (and the number of species), and the total number of assigned reads. The value in parentheses is the percent of total reads. Kraken2 provided assignments at the *Bacillus cereus* Group level, but these were counted as genus assignments.

| No. Reads | | Tool | Reads assigned to spike-in species | Reads assigned to spike-in genus | Combined | Reads assigned to other species within genus (No. Unique Species) | Total assigned reads |
|---|---|---|---|---|---|---|---|
| *B. anthracis* 2% spike-in | 6,353,624 | MTSv | 54,846 (0.86%) | 199,560 (3.1%) | 254,406 (4.0%) | 687 (51) (0.01%) | 384,021 (6.0%) |
| | | Centrifuge | 53,443 (0.84%) | 195,056 (3.07%) | 248,499 (3.91%) | 2,443 (80) (0.04%) | 1,769,907 (27.9%) |
| | | Kraken2 | 43,596 (0.69%) | 208,798 (3.29%) | 252,394 (3.97%) | 2,838 (77) (0.04%) | 963,480 (15.2%) |
| *F. tularensis* 2% spike-in | 7,778,056 | MTSv | 100,492 (1.29%) | 1,246 (0.02%) | 101,738 (1.31%) | 31 (4) (0%) | 268,019 (3.4%) |
| | | Centrifuge | 98,382 (1.26%) | 776 (0.01%) | 99,158 (1.27%) | 63 (11) (0%) | 2,045,753 (26.3%) |
| | | Kraken2 | 100,104 (1.29%) | 1,621 (0.02%) | 101,725 (1.31%) | 23 (8) (0%) | 1,009,429 (12.98%) |
| *Y. pestis* 2% spike-in | 6,397,098 | MTSv | 30,966 (0.48%) | 112,008 (1.75%) | 142,974 (2.2%) | 280 (6) (0%) | 280,577 (4.6%) |
| | | Centrifuge | 29,688 (0.46%) | 110,306 (1.72%) | 139,994 (2.19%) | 420 (14) (0.01%) | 1,651,364 (25.8%) |
| | | Kraken2 | 26,345 (0.41%) | 116,199 (1.82%) | 142,544 (2.23%) | 449 (11) (0.01%) | 857,922 (13.4%) |

large impact on the number of assignments to the spike-in species. The unfiltered Kraken2 results showed large increases (>2-fold) in 13 additional taxa in the *Bacillus* genus with a maximum of a 16-fold increase in *B. wiedmannii*. Filtering the low confidence assignments reduced this to eight taxa with more than a 2-fold increase and maximum 9-fold increase for *B. thuringiensis*. Prior to filtering, Centrifuge showed >2-fold increase in six *Bacillus* species but after filtering this increased to seven species with a maximum of a 6-fold increase. Centrifuge appeared to perform better prior to filtering but this was because it had many more assignments to *Bacillus* species in the soil background compared to the other tools and most of them were of very low quality (score < 150, hit length < 32 bp). The higher number of (albeit low quality) assignments for *Bacillus* species in the background sample masked the increase due to the spike-in prior to filtering them out. MTSv had only three *Bacillus* species with greater than a 2-fold increase with a maximum of 5-fold.

## MTSv is reasonably fast for a full-alignment algorithm and is flexible enough to run on lower memory or limited CPU environments

MTSv offers a lot of flexibility when it comes to managing the resources required for taxonomic assignment against large databases. This begins with the index building step. Unlike Centrifuge, Kraken2, and other tools, MTSv allows the indices to be split up into different sized chunks to accommodate resource limitations. For example, because the Centrifuge index must be built all at once and requires a large amount of RAM that scales with the size of the reference genome, we were restricted to using a smaller reference database in this article to allow direct comparison with Centrifuge. MTSv indices can be built using any reference database or even multiple reference databases because it breaks them into smaller chunks that require less memory to build and run. MTSv can easily

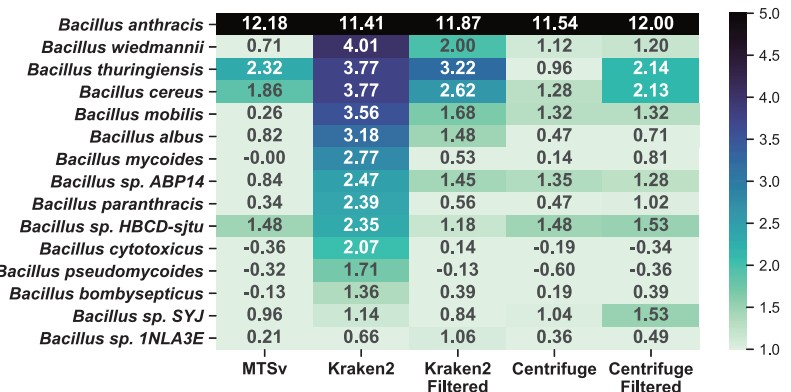

**Figure 7 MTSv showed relatively little change in the number of assignments to near-neighbor species in the spike-in sample compared to the soil background.** The heat map shows the difference (log$_2$ ratio of reads per million) in species-level assignments within the *Bacillus* genus between the soil background and the *B. anthracis* spike-in samples using MTSv, Centrifuge, and Kraken2. The heatmap shows only rows where at least one of the columns had a value greater than one indicating an increase in assignments in the spike-in compared to the soil background. A pseudocount of one was added to the sample and background counts to avoid undefined ratios.

**Table 3 Resources required to build MTSv MG-indices.** The top row of the table shows the resources required to build the MTSv indices used throughout this article. The second row shows an alternative approach that reduces the amount of memory required for building the indices by splitting the reference sequences into smaller files. The alternative method also used a higher sampling rate of the BWT occurrence array (512 *vs* 64) which resulted in a smaller index size. The last two rows show the resources required to build the data structures for Centrifuge and Kraken2.

| Tool | CPUs | Wall clock time (Avg. H:M:S per job) | Max RAM (Avg. GB per job) | Storage size on disk (*vs.* original FASTA size) |
|---|---|---|---|---|
| MTSv | 1 × 13 jobs | 01:23:07 × 13 = 18:00:13 | 251.0 | 447.3G (~3.6×) |
| MTSv (Alt.) | 1 × 66 jobs | 00:17:44 × 66 = 19:30:24 | 49.0 | 298.8G (~2.3×) |
| Centrifuge v1.0.4 | 32 × 1 job | 13:33:32 | 316.9 | 56.5G (~0.5×) |
| Kraken2 v2.1.2 | 32 × 1 job | 3:55:14 | 33.5 | 35.2G (~0.3×) |

combine the assignment results across multiple indices and additional indices can be added without needing to rebuild the entire database.

In this article, we chose to split the reference sequences among 13 different MG-indices for MTSv. Each index required 251 GB of memory and an average of 1 h 23 min to build. Alternatively for more memory limited situations, the reference sequences could be split into 66 ~2 GB files which would each require 49 GB of memory and about 18 min each to build (Table 3). Both approaches required less memory than Centrifuge's build process which required 317 GB of RAM. The storage size of the MTSv indices (using the default suffix array and occurrence array sampling intervals of 32 and 64, respectively) was ~3.6 times the size of the original reference sequences. Although this is larger than Kraken2 (~0.3×) and Centrifuge (~0.5×), storage space is arguably the least expensive and most readily available resource. The alternative example in Table 3 (MTSv alt) was built with a higher occurrence array sampling rate of 512 so the files took up less space on disk (~2.3× the original FASTA) but they required slightly longer binning times.
**Table 4 Resources required to run taxonomic assignments.** The table shows the amount of time and memory requirements to run each tool with eight threads per job. MTSv was run with 13 different jobs whereas Centrifuge and Kraken2 had only a single job each.

| Tool | CPUs | Wall clock time (Avg. H:M:S per job) | Max RAM (Avg. GB per job) |
|------|------|--------------------------------------|---------------------------|
| MTSv | 8 × 13 jobs | 29:51 × 13 = 6:28:07 | 34 |
| Centrifuge v1.0.4 | 8 × 1 job | 29:59 | 55 |
| Kraken2 v2.1.2 | 8 × 1 job | 1:11 | 29 |

Because MTSv performs full alignments, it cannot compete with the speeds of exact matching tools, but it is reasonably fast compared to traditional alignment tools. For taxonomic assignment, the average time per index was about 30 min for MTSv and each required about 34 GB of memory (Table 4). If each of the 13 indices were run simultaneously then the run time would be equal to the time for the longest index (~40 min). If each index was run in series, it would take about 6 and a half hours or about 13 times longer than Centrifuge.

## DISCUSSION

MTSv provides a good middle ground between highly sensitive traditional alignment tools that are too computationally expensive for taxonomic profiling and faster exact matching-based approaches that can be less sensitive and less precise. MTSv was not designed to compete with tools like Kraken2 or Centrifuge, but rather fill a role when higher accuracy and confidence is required from taxonomic assignments and as an alternative to traditional alignment tools used in other metagenomic analysis pipelines (*e.g.*, Metalign (*LaPierre et al., 2020*)). Here we show that our full alignment approach (1) increases precision and reduces false positive assignments, (2) better captures assignments for taxa that share low similarity with reference sequences, and (3) provides high confidence detection of pathogens while avoiding off-target assignments to near-neighbor species.

We compared MTSv to two traditional alignment tools to illustrate the issues that arise from off-label use for taxonomic assignment against multi-species reference databases. Bowtie2 is one of the most popular rapid read alignment tools and one of the first to use an FM-index data structure to rapidly look up seeds to extend for full alignments (*Langmead & Salzberg, 2012*; *Alser et al., 2021*). Minimap2 is a state-of-the-art hash-based alignment tool that is faster than mainstream short-read aligners with comparable accuracy (*Li, 2018*; *Alser et al., 2021*). In our small example, we used a collection of genomes that were dominated by a single species and included species that were closely related. Although this was a toy example to demonstrate a point, it is not far off from reality. Many clinically or economically important species are overrepresented in reference databases and other closely related, yet less relevant species may only have a single complete genome. When using traditional alignment tools, accurate assignment can rely on finding an alignment to a single genome among many before the level of search effort is exhausted. If other valid alignments are found among sequences for closely

related taxa, the search effort is more likely to be exhausted before the correct alignment is found. This leads to a bias for assignments to more abundant genomes and noisy results, especially when extended beyond our toy example.

One approach to overcome the bias of traditional alignment tools is to increase their search effort to include all valid alignments. Unfortunately, this made the tools very inefficient in terms of speed and, for Minimap2, memory usage, even on a small toy dataset. Attempts to scale traditional read alignment tools to realistic sized metagenomic samples with larger more complex reference databases are sometimes unsuccessful (*Jaillard et al., 2016*) and increasing the search effort will only make this more impractical. An alternative approach could involve building separate indices for each taxon. Unfortunately, this is also impractical as it would require maintaining and updating over 4,700 separate indices for just the bacterial reference sequence database used here. MTSv was designed specifically to overcome these challenges because it was purpose built to handle taxonomic assignment of metagenomic reads against multi-species reference indices. MTSv does an exhaustive unbiased search for alignments across all reference sequences but it is taxon-aware so once it finds a valid alignment for a taxon, no additional alignments are attempted against any reference for that taxon. This prevents missed assignments while providing large speedups.

We compared MTSv to two taxonomic assignment tools, Centrifuge and Kraken2, to demonstrate the situations where MTSv may provide superior performance. Using data with known read assignments, MTSv had high precision and sensitivity at each taxonomic level. Consequently, MTSv reduced the number of false positive species identified by thousands. The false bins introduced with Centrifuge and Kraken2 were larger and therefore are more likely to be treated as a positive signal. Noisy results interfere with true positive signals and make it difficult to detect low abundance taxa with confidence. MTSv reports only high-quality alignments (with >87% similarity with the reference) and reduces the number of false positives thereby increasing overall confidence in the assignment results.

Exact match approaches underperform when sequences diverge from reference genomes. Differences between a read and a reference sequence can occur due to sequencing errors, DNA degradation (*e.g.*, ancient DNA; *Prys-Jones et al., 2022*), or taxon diversity that is not captured by existing databases. These differences can rapidly decay the potential for exact matches by interrupting stretches of sequence identity and modifying parameters to adjust for this (*e.g.*, reducing the fixed k-mer size in Kraken2) often sacrifices resolution. By performing a full alignment, MTSv can account for mismatches and assign reads that would be missed by other tools and the alignment sensitivity can be modified easily to capture more divergent sequences. With sensitive parameters, MTSv found 80% of assignments for reads that diverged up to 15% from reference sequences, whereas Kraken2 and Centrifuge found less than 20%. Increasing sensitivity also increases runtime but this can be mitigated by running sensitive mode only on reads that could not be assigned using the default mode.

The assignments missed due to sequence divergence are likely to influence downstream species abundance estimates. Both Centrifuge and Kraken2 (with the help of Bracken

(*Lu et al., 2017*)) can provide species abundance estimates based on the number of reads assigned to each taxon. Instead of using only species level assignments, Bracken improves abundance estimates by redistributing reads assigned at the genus level or higher among the species level counts. However, it cannot correct for missing or false species-level assignments. We showed that when an organism diverges from reference sequences, Kraken2 and Centrifuge completely miss many assignments which reduces the read count for the species and cannot be corrected using Bracken. For this reason, the abundance estimates are highly dependent on the accuracy of species level assignments. Minor fluctuations in the number of assigned reads will likely have little effect but our results show that divergent genomes caused large shifts in read abundance rankings among species.

K-mer based approaches like Kraken2 must strike the right balance between sensitivity and precision when picking a k-mer size. With longer k-mers, fewer exact matches are likely due to sequencing errors or biological variation. However, as k-mer size decreases, there are fewer unique k-mers per taxon and the resolution suffers (*Lu et al., 2017*; *Wood & Salzberg, 2014*; *Wood, Lu & Langmead, 2019*). This is particularly challenging when trying to detect specific pathogenic species in metagenomic samples when closely related species share many identical k-mers. Low coverage may result in missing the k-mers that differentiate between the taxa and sequencing errors can easily cause false positive assignments. For example, the anthrax pathogen, *Bacillus anthracis*, shares >99% average nucleotide identity with its closest relatives in the *Bacillus cereus* group that includes ubiquitous soil bacteria like *Bacillus cereus* and *Bacillus thuringiensis* (*Helgason et al., 2000*; *Rasko et al., 2005*). This high degree of similarity led to the re-analysis of a sensational 2015 study that identified the alarming presence of *B. anthracis* in metagenomic samples collected from New York City subways (*Ackelsberg et al., 2015*; *Afshinnekoo et al., 2015a, 2015b*). The re-analysis revealed that the detection of *B. anthracis* was likely an artifact of high levels of sequence overlap with other species. We show here that Kraken2 was susceptible to misclassifying reads within the *Bacillus* genus even after filtering out low confidence assignments. Even tools like KrakenUniq (*Breitwieser, Baker & Salzberg, 2018*) which were designed specifically to avoid false positives in pathogen detection has been prone to falsely predict the presence of *B. anthracis* when other *Bacillus cereus* Group species are present (*Petit et al., 2018*). We are working on developing similar methods that will provide confidence estimates for the true presence of taxa in metagenomic samples, but out of the box, MTSv offered more high-quality assignments in our pathogen detection experiments without creating a large cloud of additional assignments to near neighbor species. This is highly valuable for higher-stakes pathogen detection situations (*e.g.*, analyzing clinical or biothreat samples).

Our goal when developing MTSv was to provide an alignment algorithm that was specifically designed for analyzing metagenomic data. Although traditional sequence alignment tools are very efficient, they must perform exhaustive searches to avoid missing alignments using multi-species reference databases which become computationally expensive at scale. In metagenomics, it is less important to identify all the possible alignments and more important to know that an alignment of sufficient quality was found

among the reference sequences for a given taxon. MTSv performs a taxon-aware exhaustive search but skips references for TaxIDs where a valid alignment has been found. This allows MTSv to provide full alignment-based taxonomic assignments at speeds that are feasible for metagenomic data. MTSv is more flexible than other existing tools when it comes to modifying the pipeline to work within different computational restrictions. For example, with Centrifuge, the FM-index must be built all at once which requires large amounts of memory that scales with the size of the reference database (we did not have the resources to build anything larger than the bacteria-only complete genomes database). The Centrifuge developers do provide prebuilt indices but that limits control of updates and customization. MTSv indices, on the other hand, can be built in smaller units that suit the user's computing resources and additional reference sequences can be added without needing to rebuild the complete data structure. Although MTSv cannot compete with the assignment speeds of the exact matching approaches, it offers several key advantages while still being faster than other traditional alignment tools.

## CONCLUSIONS

In conclusion, MTSv provides a novel and relatively fast algorithm for taxonomic classification based on full alignment of nucleotide sequences. It offers greater precision, and confidence in assignments than exact-matching tools which is required in some situations. It offers flexibility to fit different computational needs and can easily expand to accommodate rapidly growing reference databases.

## ACKNOWLEDGEMENTS

The authors are grateful for the programming work provided by Adam Perry, and for the support and guidance of Johannes Köster and other contributors to the Rust-Bio library. The authors also want to acknowledge the support from the Northern Arizona University high-performance computing staff.

### Funding

This work has been supported by the Department of Homeland Security Grant (HSHQDC-17-C-B0008/BAA14-003). Software development, benchmarking, and analysis was performed on Northern Arizona University's Monsoon computing cluster, funded by Arizona's Technology and Research Initiative Fund. The funders had no role in study design, data collection and analysis, decision to publish, or preparation of the manuscript.

### Grant Disclosures

The following grant information was disclosed by the authors:
Department of Homeland Security Grant: HSHQDC-17-C-B0008/BAA14-003.
Northern Arizona University's Monsoon Computing Cluster.
Arizona's Technology and Research Initiative Fund.

## Competing Interests

The authors declare that they have no competing interests.

## Author Contributions

- Tara N. Furstenau conceived and designed the experiments, performed the experiments, analyzed the data, prepared figures and/or tables, authored or reviewed drafts of the article, and approved the final draft.
- Tsosie Schneider performed the experiments, analyzed the data, authored or reviewed drafts of the article, and approved the final draft.
- Isaac Shaffer performed the experiments, analyzed the data, authored or reviewed drafts of the article, and approved the final draft.
- Adam J. Vazquez performed the experiments, authored or reviewed drafts of the article, and approved the final draft.
- Jason Sahl conceived and designed the experiments, performed the experiments, authored or reviewed drafts of the article, and approved the final draft.
- Viacheslav Fofanov conceived and designed the experiments, performed the experiments, analyzed the data, authored or reviewed drafts of the article, and approved the final draft.

## DNA Deposition

The following information was supplied regarding the deposition of DNA sequences:

The soil spike-in sequences are available at NCBI's Sequence Read Archive: SAMN31055686 (*B. anthracis* mixture), SAMN31055687 (*Y. pestis* mixture), SAMN31055688 (*F. tularensis* mixture), and SAMN31055689 (soil background).

## Data Availability

The code is available at GitHub: https://github.com/FofanovLab/mtsv_tools and Zenodo: Furstenau, Tara N., Schneider, Tsosie, Perry, Adam, & Fofanov, Viacheslav. (2022). FofanovLab/mtsv_tools: v2.0.1 (2.0.1). Zenodo. https://doi.org/10.5281/zenodo.6369369.

The raw data, and other analysis scripts are available at Zenodo: Furstenau, Tara N., Schneider, Tsosie, Shaffer, Isaac, Vazquez, Adam, Sahl, Jason, & Fofanov, Viacheslav. (2022). MTSv: Rapid alignment-based taxonomic classification and high-confidence metagenomic analysis. https://doi.org/10.5281/zenodo.6899690.

## Supplemental Information

Supplemental information for this article can be found online at http://dx.doi.org/10.7717/peerj.14292#supplemental-information.

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
