# Peer review of "MTSv: rapid alignment-based taxonomic classification and high-confidence metagenomic analysis"

_PeerJ, doi:10.7717/peerj.14292_

## Round 0.1 · original submission · Major Revisions

We received three reviews that all recommended major revisions. These reviewers were detailed and constructive. The consensus is that MTSv has a niche in metagenomic analysis and it is a valuable contribution. I agree with this recommendation and the general suggestion that MTSv should be compared to other alignment tools, at the very least in Introduction and Discussion. I also agree that the computational speed benefits of MTSv, relative to BowTie, should be quantified in Results.

I have a number of suggestions for format and style for your consideration.

line 18 -22. This is wordy. Replace “Modern exact … reference sequences” with “Exact matching approaches can rapidly assign taxonomy and summarize the composition of microbial communities but they sacrifice accuracy and can lead to false positives. Full alignment tools provide higher confidence assignments and can assign sequences from genomes that diverge from reference sequences; however, full alignment tools are computationally intensive.”
Line 22 – 23. Replace “We developed MTSv …. Smith-Waterman alignment.” with “To address this, we designed MTSv specifically for taxonomic assignment in metagenomic analysis. This full-alignment tool implements an FM-index assisted q-gram filter and SIMD accelerated Smith-Waterman alignment runs to run efficiently.”
Line 28-29. Replace “it is reasonably fast. We show that MTSv … Kraken2” with “it is reasonably fast and has higher precision than popular exact matching approaches”
Line 29. Replace “… it is better able to…” with “it can”
Line 31-32. Replace “MTSv also provides higher confidence in pathogen detection and reduces off-target assignments ….” With “MTSv provides a tool for high confidence pathogen detection and low off-target assignments…”
Line 37. I think advances in isolation methods and high-throughput microbial identification by MALDI-TOF has disrupted the dogma that only a “few individual members” can be cultured. For example, Didier Raoult’s lab alone has cultured the majority of gut microbes (Lagier 2016 Nat Micro 1:16203)
Line 66. Biocontamination is one word.
Line 230 – 236. Start each paragraph in Results with a statement of the result presented in a particular figure or table, not with a restatement of Methods. Also, line 233-234 is a figure legend not a results.
Line 266-275. As above, start paragraphs with a statement of a Result.
Line 269. Delete “will” in “will tend”
Line 283. Replace “seemed to be highly dependent” with “depended”
Line 287. Delete “The” in “The read”
Line 290. As above, start with a topic sentence that states a result.
Line 317. This phrase is a figure legend not a result “…Fig. 5 shows the log2 ratio of reads per million… soil background.”
Line 389. Delete “As a result”
Line 433. Here and throughout, write actively. Replace “was able to clearly detect” with “detected”
Line 436. As above, revise to “MTSv made more..”
Line 474 and 492. Be consistent in format of references. For example, “.. Lack of Evidence for Plague or Anthrax on the New York City Subway…” caps first letter in most words. I think the correct format is only proper nouns “Microbes as forensic indicators.”
Line 492. Format for journal tiles is “Tropical Biomedicine.” See also line 570, 627, 630, 642..
Line 508. This citation is missing a source. I think it is “Digital SRC Research Report.”
Line 527. Italicize genus and species. See also 623.
Line 576. Source is missing for Kühl.
Line 624. Why star?
Line 633. Page number is missing.

Reviewer 1 ·

Basic reporting

The article is clearly written and easy to read. There is sufficient background provided and extensive literature references. The authors very clearly delineate the purpose, strengths, and weaknesses of the tool they propose.

Experimental design

The research question (i.e. how well does the proposed method, MTSv, classify metagenomic reads) is clearly stated and the authors note the drawback of current alignment-based approaches to metagenomic classification, thus making clear how their approach fills a knowledge gap in the field.
The experimental design is quite rigorous: typically in these sort of methods publications, time is spent analyzing simulated or publicly accessible data on which to benchmark an approach and compare it to other approaches. Here, in addition to using this sort of benchmark data, the authors design an experiment in which a pathogen is spiked into a soil sample, thus creating a gold-standard, real-world data set on which to test MTSv and other approaches to the pathogen detection problem. I applaud the authors for going to this effort when others may have instead used an in-silico spike-in instead.

Validity of the findings

As for availability and reproducibility, I was able to install and use the associated tool MTSv. However, I would suggest that the authors put a link to the software (https://github.com/FofanovLab/mtsv_tools) in the main manuscript itself to make it easier for users to find it.

The conclusions are well stated and mostly limited to supporting results. However, I did find it odd that the authors repeatedly claimed that MTSv improves upon alignment tools like Bowtie2 (see lines 374-376). However, there is little to no actual comparison to Bowtie2 or other such alignment tools, rather the focus was on K-mer based methods such as Kraken2.The only mention of an experiment involving Bowtie2 is in lines 381-385 where a simple speed-based comparison is performed. Given the author's focus on improving alignment-based methods (eg. mentioning how traditional alignment tools are lacking in the very first sentence of the abstract). In addition, the authors appear to either be unaware of, or omit, a discussion (or comparison to) recent methods that have been introduced that facilitate improved metagenomic read alignment. In particular, Metalign (https://doi.org/10.1186/s13059-020-02159-0) and Sourmash (multiple pubs, code here: https://github.com/sourmash-bio/sourmash) which both use a hashing-based approach to filter out reference genomes that are likely not in a given sample, thus circumventing the issues mentioned by the authors in places such as the paragraph starting in line 374 and 440. Additionally, modern aligners (such a minimap2, and even BLASTn/x) allow for "best hit" settings so the aligner will ignore multiple mappings, but just return the best alignment. It seems appropriate to compare:
1) MTSv to alignment methods like Bowtie2, minimap2, and Diamond in terms of accuracy while
2) using setting for the aligners that either a) return only the best alignment or b) use strict, instead of default, settings that are more appropriate for metagenomic analysis (eg. https://academic.oup.com/bioinformatics/article/32/12/1779/1743377), and
3) include a discussion or comparison to alternate approaches to improving metagenomic mapping (like Metalign and Sourmash mentioned previously)

Additional comments

Minor comments:
line 17: "it is no longer computationally feasible to use traditional alignment tools for taxonomic classification" -> this is not technically true. MetaPhlAn1/2/3 and mOTUs use traditional alignment for taxonomic classification (technically: taxonomic profiling) and achieve this by using a smaller reference database of clade specific marker genes. Metalign and Sourmash accomplish this by using a pre-alignment filtering step. The authors should clarify that they intent "traditional alignment to large databases of whole genome sequences for taxonomic classification is infeasible" or the like.

Lines 273-274 what is the definition of "similarities" here? Average nucleotide identity, or something else? Clarification needed.

Lines 441-445: Some aligners (eg. Bowtie2 & minimap2) have setting that allow one to circumvent the exhaustive search for alignments. For example, setting the --best_hit_overhang in BLAST or the -k setting in Bowtie2 can speed their performance. Clarification should be made here that the authors are discussing default settings only. Better yet, the authors could discuss how these aligners were not originally intended for taxonomic classification of metagenomic samples, so their default settings are not appropriate, but rather need to be modified to allow for better performance.

Reviewer 2 ·

Basic reporting

The manuscript by Furstenau et al. presents a novel tool for the taxonomic classification of metagenomic reads.

Some of the points of force of the presented approach, named MTSv, are the fast running time and the low memory requirement, although it seems it is not present a comparison with the competitor tools in these two aspects.

Another important aspect here stressed by the authors is the unfeasibility of traditional alignment tools considering the ever-increasing size of metagenomic datasets. Although this might be true that it could be a bottleneck, I think it would be interesting and important to compare MTSv with alignment-based approaches, given that the chosen competitors are only kmer-based ones.

Experimental design

In the comparisons, it would be nice to also see a qualitative representation to show that indeed MTSv improves on the actual taxonomic representation of microbiome samples.
Also, the standard output from the Opal evaluation as used in the CAMI2 challenge would help to better interpret and relate the new results with also other competitor tools, even though they are not included here.

Validity of the findings

no comment

Reviewer 3 ·

Basic reporting

a) Although the paper references Centrifuge and Kraken2, there are a number of other relevant tools which should at least be mentioned. Specifically, mmSeqs, Bracken, and KrakenUniq should be mentioned in the background.

Experimental design

MTSv as a metagenomics tool seems to fall between Bowtie2 and the Kraken/Centrifuge type of tools. However, some details have been left out and some methods should be reconsidered.

a) Although authors detail how they ran Kraken/Centrifuge, they do not explain how they ran/tested Bowtie2 despite providing some comparison in the discussion section

b) The comparison of MTSv using 13 different databases vs. Centrifuge/Kraken2 using 1 database each is a bit misleading when reporting on runtime/RAM requirements. This should be clarified.

c) Centrifuge and Kraken can be built using the same k-mer lengths. Authors should consider comparisons using the same k-mer lengths.

d) KrakenUniq (which also has been implemented in Kraken2) has not been referenced in the paper, nor included in the pathogen detection section. KrakenUniq itself is built to report on the number of kmers supported by read classifications. This is crucial in pathogen detection.

e) Taxonomic classification tools do not necessarily provide species-level assignments. Abundance estimation tools do. Measuring the accuracy of Kraken and Centrifuge at a species level without considering Bracken (at least for Kraken) is misleading, as was detailed in the Bracken paper.

Validity of the findings

a) True Positives (TP), False Positives (FP) etc should be clearly defined, especially in the case of classification tools. As stated in the Kraken2 paper, because classification tools may assign a read to higher in the taxonomy, the standard definitions for accuracy must be modified to account for reads assigned to a correct parent taxon. (For example, it is correct to say a read that is found in two different Mycobacterium species should be labeled only as Mycobacterium, not as one of the individual species)

b) The claim on Line 385 should be carefully supported "in cases where many similar sequences were present in the index, we projected that Bowtie2 would take over 2,000 hours to complete". There is no data/examples/calculations provided to support this claim.

---

## Round 0.2 · Minor Revisions

We received two re-reviews of this version 1 and the consensus is that manuscript is scientifically sound and should be published. There are some minor comments from one review that should be addressed and I have a few comments below. If these comments are addressed adequately the manuscript will accepted without another round of external review.

- Line 304 I couldn't find an SRA project for the pathogen spike in experiment. It would also help if the qPCR methods and results mentioned in lines 304-313 are described in supplemental materials
- line 379. I think this paragraph needs a better topic sentence. Starting with the phrase "Figure... shows that" directs the reader away before they are informed what they should look for. I suggest starting with something like "The number of reads assigned accurately by Centrifuge and Kraken2 depended on..."

Reviewer 1 ·

Basic reporting

No comment, see point 4

Experimental design

No comment, see point 4

Validity of the findings

No comment, see point 4

Additional comments

With the inclusion of a comparison to Bowtie2 and Minimap2 (along with a discussion about the “off-label” use of these tools), the authors’ claim that MTSv improves upon such methods is much better supported. It was particularly convincing to see the resource comparison between “best hit” and exhaustive settings for bowtie2 and Minimap2. This helps put into context the results in Table 1 about the number of misses assignments for the different aligner settings. With the inclusion of the new results, the authors very convincingly argue that traditional read alignment tools were not designed for, not perform well, at the task of metagenomic taxonomic assignment, and the tool they introduce performs this task in a much more accurate way.

The background and context have been significantly improved with a more expansive discussion about tools and approaches in this space and how they compare and contrast to MTSv.

The authors have included a link to the GitHub page which hosts their software and I was successfully able to install and run their tool.

All other issues that I have raised have been addressed by the authors.

Reviewer 2 ·

Basic reporting

The manuscript greatly improved from its previous version. Although it is not clear to me which is the final version. The document with tracked changes does not match the version available in PDF and the clean one. So, below are some points with line numbers referring to the tracked changes document.

Line 54: "meethods" --> methods
Line 239: "Supplemental Table XX", please specify
Lines 240-241: "(Supplemental Script XX)", please specify
Lines 316, 322, 327, 412, 433, 437: formulas are empty ?

Experimental design

The experimental design is good as more comparisons were added in the revised version.

Validity of the findings

no comment

---

## Round 0.3 · accepted · Accept

I enjoyed managing this manuscript.

Regards,

Michael